# Parameterization of intraoperative human microelectrode recordings: Linking action potential morphology to brain anatomy

**Matthew R. Baker** [iD] [1¤*], **Bryan T. Klassen** [2], **Michael A. Jensen** [1], **Gabriela Ojeda Valencia** [3], **Hossein Heydari** [4,5], **Nuri F. Ince** [1,4,5], **Klaus-Robert Müller** [6,7,8], **Kai J. Miller** [1,4,9]

**1** Department of Neurosurgery, Mayo Clinic, Rochester, Minnesota, United States of America, **2** Department of Neurology, Mayo Clinic, Rochester, Minnesota, United States of America, **3** Graduate School of Biomedical Sciences, Mayo Clinic, Rochester, Minnesota, United States of America, **4** Department of Physiology and Biomedical Engineering, Mayo Clinic, Rochester, Minnesota, United States of America, **5** Department of Bioinformatics and Computational Biology, University of Minnesota, Minneapolis, Minnesota, United States of America, **6** Department of Computer Science, TU Berlin, Berlin, Germany, **7** Dept of Artificial Intelligence, Korea University, Seoul, Republic of Korea, **8** Max Planck Institute for Informatics, Saarbrücken, Germany, **9** Department of Pediatrics, Mayo Clinic, Rochester, Minnesota, United States of America

¤ Current address: Department of Neurosurgery, Mayo Clinic, Rochester, Minnesota, United States of America

* Baker.Matthew@mayo.edu

## Abstract

Deep brain stimulation (DBS) is a targeted manipulation of brain circuitry to treat neurological and neuropsychiatric conditions. Optimal DBS lead placement is essential for treatment efficacy. Current targeting practice is based on preoperative and intraoperative brain imaging, intraoperative electrophysiology, and stimulation mapping. Electrophysiological mapping using extracellular microelectrode recordings aids in identifying functional subdomains, anatomical boundaries, and disease-correlated physiology. The shape of single-unit action potentials may differ due to different biophysical properties between cell-types and brain regions. Here, we describe a technique to parameterize the structure and duration of sorted spike units using a novel algorithmic approach based on canonical response parameterization, and illustrate how it may be used on DBS microelectrode recordings. Isolated spike shapes are parameterized then compared using a spike similarity metric and grouped by hierarchical clustering. When spike morphology is associated with anatomy, we find regional clustering in the human globus pallidus. This method is widely applicable for spike removal and single-unit characterization and could be integrated into intraoperative array-based technologies to enhance targeting and clinical outcomes in DBS lead placement.

**Data availability statement:** All code to implement this technique along with sample data to reproduce figures and analyses are publicly available for use without restriction (other than attribution) at: https://osf.io/qcgeh/.

**Funding:** BTK and KJM were supported by the MN partnership grant for biotechnology and medical genomics (MNP2142). KJM was supported by the Van Wagenen Fellowship, the Brain and Behavior Research Foundation with a NARSAD Young Investigator Grant, and the Foundation For OCD Research. KRM was supported in part by the German Ministry for Education and Research (BMBF) under Grants 01IS14013A-E, 01GQ1115, 01GQ0850, 01IS18025A, 031L0207D, 01IS18037A as well as Berlin Institute for the Foundations of Learning and Data (BIFOLD). This work was also supported by the National Institutes of Health (NIH) NCATS CTSA KL2 TR002379 (KJM), NINDS U01-NS128612 (KJM), NINDS R01NS124650 (NFI), and NINDS R01NS112497 (NFI). Manuscript contents are solely the responsibility of the authors and do not necessarily represent the official views of the NIH. The funders had no role in study design, data collection and analysis, decision to publish, or preparation of the manuscript.

**Competing interests:** The authors have declared that no competing interests exist.

## Author summary

We developed a new algorithmic approach to capture the structure of single neuron electrophysiology from microelectrode recordings during deep brain stimulation surgery. This method has broad applications for spike characterization, clustering, removal from the background field potential, and comparison across recording sites. Using this technique, we illustrate regional clustering of spike morphologies in the human globus pallidus.

## Introduction

The microcircuitry of the human brain is made up of diverse cell-types, each with distinct connections, morphology, and function [1,2]. These biophysical properties are associated with a significant variation in action potential shape, duration, and firing pattern which can facilitate identification of neuronal cell-types. However, in humans, this process is particularly challenging due to reliance on extracellular recordings with clinical-grade tools, as opposed to the many pharmacologic, transgenic, and experimental-grade electrophysiology tools available in animal models. Traditional approaches to characterize and classify the morphology of extracellular action potentials involve selecting features such as peak-to-trough duration, spike width, firing rates, and others, which may not capture the complete spike structure and rely on arbitrary selection of features.

This work proposes a novel algorithmic approach to discover the structure and duration of sorted spike units, and parameterize individual action potentials in a mathematically principled way. Our approach is built on canonical response parameterization [3], which was previously developed for parameterizing evoked-responses from single-pulse electrical stimulation. This allows for: 1) discovering structure of action potentials and removing them from background signal, 2) quantifying intuitive metrics for each spike unit, such as signal-to-noise, variance explained, and magnitude, and 3) facilitating shape comparisons from different recording sites across resolved spike intervals. We demonstrate the application of our novel approach to human microelectrode recordings in patients undergoing deep brain stimulation (DBS) surgery.

DBS is a direct therapeutic approach to brain circuit disorders, providing sophisticated treatment options for a wide range of neurological and neuropsychiatric conditions [4–6]. The accurate placement of permanent leads within the planned structure or substructure is critical for the efficacy of DBS treatment. Current practice for DBS targeting is based on a combination of anatomical borders visible on pre-operative magnetic resonance imaging (MRI) and functional targeting through intraoperative stimulation testing and electrophysiological mapping [7,8]. However, patient-specific variation in brain circuits limits the effectiveness of using imaging alone [9]. Functional electrophysiology mapping is accomplished through extracellular intraoperative microelectrode recordings (MER). MER involves inserting a single-channel microelectrode into the computed brain target to identify neuronal structure boundaries and disease-correlated physiology. The microelectrode is advanced manually into the deep brain in small, typically sub-millimeter steps, stopping at each step to record neural activity in the region of the electrode tip.

While MER targeting has identified common firing patterns and "cell types" associated with DBS target regions [10], it can often be subjective, time-consuming, and require significant expertise to interpret neural activity. Beyond firing rates and patterns, which greatly vary between patients and as a function of disease and anesthesia states [11–13], characterizing the

shape of action potentials may provide useful information for identification of distinct cell-types, anatomical borders, and electrode implant locations [14]. This algorithmic approach could serve as a valuable tool for intraoperative electrophysiological mapping, as well as a more general tool for parameterizing the structure and duration of single-unit data and it's relationship to field potentials.

## Materials and methods

### Ethics statement

All patients gave written consent to participate in a research protocol during the surgery. Mayo Clinic's internal review board approved the study and the consent process (IRB no. 19-009878).

### Patients and surgical implantation

Microelectrode data were analyzed from 3 patients receiving DBS electrode implantation of the internal globus pallidus (GPi) or ventral intermediate nucleus (VIM) of the thalamus for treatment of Parkinson's Disease and Essential Tremor, respectively. Stereotactic planning was performed using the StealthStation S8 surgical navigation system (Medtronic, Minneapolis, Minnesota). The stereotactic plan was executed utilizing the Leksell G frame system (Elekta, Stockholm, Sweden).

### Electrophysiological recordings and lead localization

A cannula was passed along the planned trajectory to 15 mm above the target. Next, a micro-electrode (0.5–1 MΩ platinum–iridium; FHC, Bowdoin, ME) was slowly advanced from the tip of the cannula to just past the planned target. Raw voltage was measured from the micro-electrode, referenced to the cannula, and sampled at 44 kHz using a NeurOmega MER system (Alpha Omega, Nof HaGalil, Israel). Fig 1A–1B shows a representative schematic for single-unit MER targeting the GPi. Serial recordings were obtained at multiple sites along the trajectory while the patient remained at rest. No somatosensory or visual evoked potential or micro/macro-stimulation testing were included in the analyzed data. Patient's may have had light dexmedetomidine sedation at the beginning of the surgery, but would have been discontinued at least 15 minutes prior to the start of MER. No additional CNS-active agents were administered during the recordings. At each site, recording commenced 2 seconds after the electrode reached the designated position and continued for approximately 30 seconds. Typically, recordings were taken every 0.5–1 mm.

The site of each recording relative to the final lead position was known, enabling its location on the patient's MRI as follows: First, a postoperative CT scan was co-registered to pre-operative MRI sequences using a normalized mutual information approach implemented in SPM12. Then, the lead artifacts were manually identified, and MRI sequences were resliced in plane with the lead artifact using custom Matlab code (Fig 1C–1D) [15]. The offset of each recording site from the final lead position was applied, and recording sites were plotted on T1-weighted, T2-weighted, and Fluid-attenuated Gradient echo with Adiabatic T1 Inversion Recovery (FGATIR) MRI sequences. MRI borders of the GPi and surrounding structures were visually identified and outlined in the figures by MRB and KJM (Fig 1E).

### Spike sorting

Raw voltage traces were filtered from 300 Hz to 9 kHz and imported into Plexon Offline Sorter (Plexon inc., Dallas, Texas, United States of America). Spike sorting was broken into

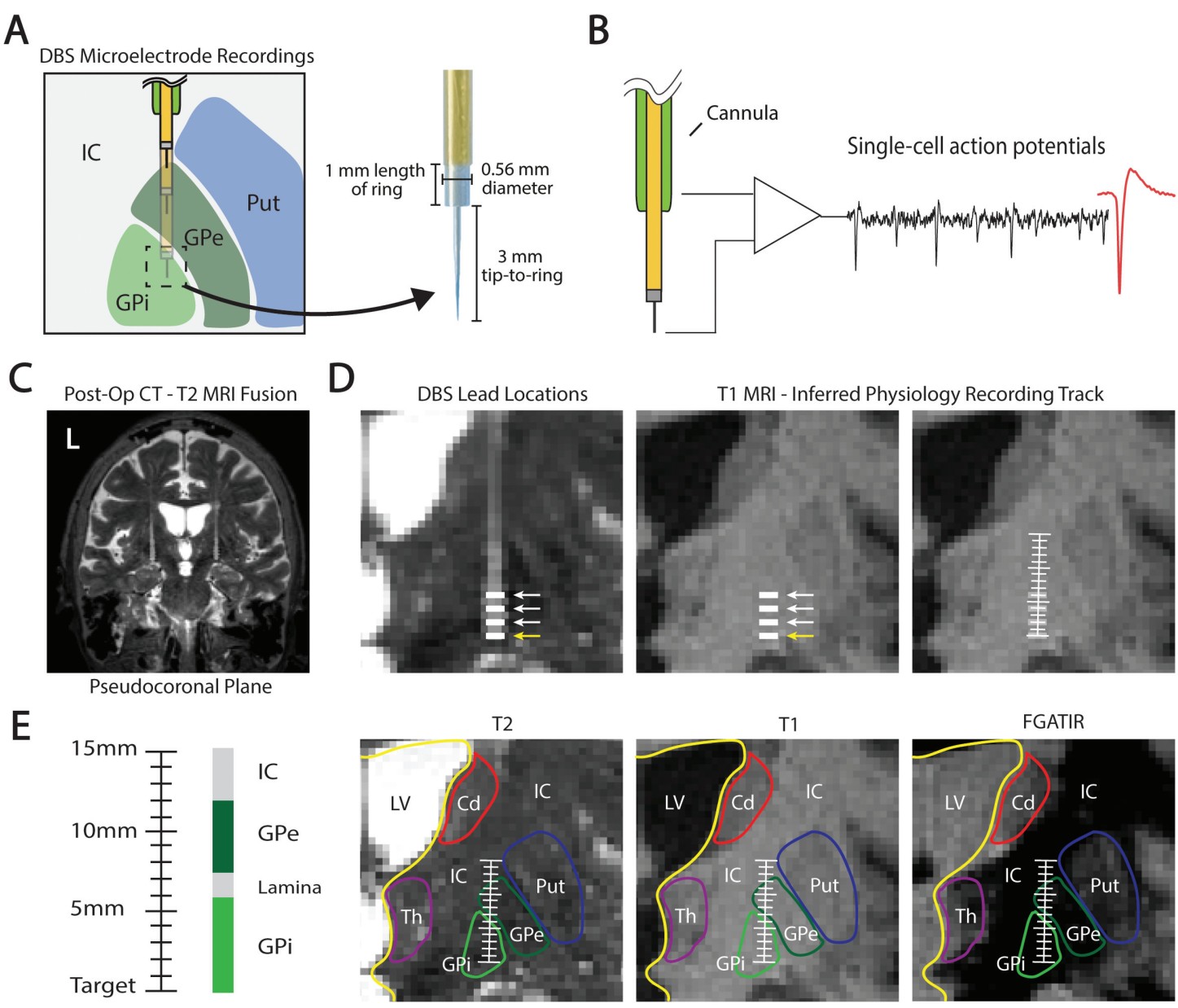

**Fig 1. GPi DBS microelectrode recordings.** (**A**) Diagram of serial microelectrode recordings for DBS implantation surgery and microelectrode dimensions. (**B**) Raw voltage is recorded from the microelectrode tip, referenced to the cannula, and filtered from 300 Hz to 9 kHz to analyze single units. (**C**) Microelectrode recording position was determined by co-registration of the post-surgical CT and pre-surgical MRI, and resliced in plane with the DBS lead position. (**D**) DBS lead contacts were identified and used to infer the microelectrode recording track. Yellow arrow indicates target depth. (**E**) T2, T1, and FGATIR MRI series were overlaid to segment GPi and surrounding anatomical borders. GPi = Internal globus pallidus; GPe = External globus pallidus; Put = Putamen; IC = internal capsule; Cd = Caudate; Th = Thalamus; LV = Lateral ventricle.

three steps: preprocessing, waveform detection, and feature-based clustering. In the pre-processing step, non-physiological events or perturbations were identified by visual inspection and removed. The spike detection threshold was set to -4 standard deviations of the mean peak heights histogram [16]. After waveform detection, 0.5ms before and 1.5ms after

threshold crossing were extracted resulting in a total possible spike window of 2ms. During detection, waveforms were aligned to the largest peak following threshold crossing.

For spike sorting, we used template match sorting, a supervised clustering method with the number of clusters manually predefined [17]. Templates were selected based on the first two principal components of the detected waveforms. For each waveform, the algorithm (1) calculates the sum-of-squares differences from this waveform to all the templates, (2) finds the template with the minimum sum-of-squares difference, and (3) if the minimum difference is less than the fit tolerance for this template, the waveform is assigned to this template's unit. The fit tolerance was set to 5 (measured in units of 0.01% of the maximum A/D value). Clusters were evaluated visually and by the refractory period violations, which indicates the percentage of spikes within a sorted unit that have an inter-spike interval less than 2ms, indicative of the intrinsic physiological firing limits of a neuron. Clusters were refined if the percentage of refractory period violations were over 2% by adding an additional cluster, increasing the stringency of the spike detection threshold, or adjusting the fit tolerance.

## Spike parameterization

Intraoperative microelectrode recordings provide clinical utility by qualitatively observing different types of neuronal firing along the physiological recording track (Fig 2A). In GPi DBS implant surgeries, the identification of neurons such as bursting and pausing cells in the GPe or border cells and irregular firing neurons in the GPi provide anatomical information that aids in decision-making for final lead placement (Fig 2B) [10]. However, the process can often be subjective, time-consuming, and require significant expertise to interpret neural activity. Thus a more automated mathematical characterization of spikes from MER could aid in identification of putative cell-types and anatomical boundaries, reduce DBS lead placement error, and improve clinical outcomes for patients.

Sorted spike units from intraoperative microelectrode recordings were characterized using canonical response parameterization (CRP) [3]. This machine learning technique was developed to discover the structure and duration of single-pulse electrical stimulation data, and is modified here for sorted spike units (Fig 3A). Individual spikes are projected onto its mean unit waveform using a semi-normalized dot product. The length of time points from the global extremum following threshold crossing (time 0) is varied to obtain a temporal profile of structural significance, with the maximum projection magnitude identifying the spike end time ($\tau_R$). Next, the process is repeated in the reverse direction starting from $\tau_R$. The new peak projection magnitude identifies the spike start time ($\tau_i$). Thus, identifying a significant spike window from $\tau_i$ to $\tau_R$, outside of which the voltage is statistically unreliable (Fig 3B).

Each individual spike ($k$) is represented as a projection of the sorted unit ($n$) shape $C(t)$, scaled by a scalar $\alpha_k$ over the spike window $t'$ (from $\tau_i$ to $\tau_R$) at spike time $\tau_k$, with a residual local field potential (LFP) $V_o(t)$ (Fig 3C–3D):

$$V(t) = \sum_k \alpha_k C_{n(k)}(t' + \tau_k) + V_o(t)$$

Knowing $\alpha_k$, we can quantify the residual LFP after regressing out the sorted unit shapes of $C_n(k)$:

$$V_o(t) = V(t) - \sum_k \alpha_k C_{n(k)}(t' + \tau_k)$$

Note, by LFP, we refer to an encompassing term for the non-spiking component of the extracellular voltage signal, which includes both low-frequency narrow-band oscillations

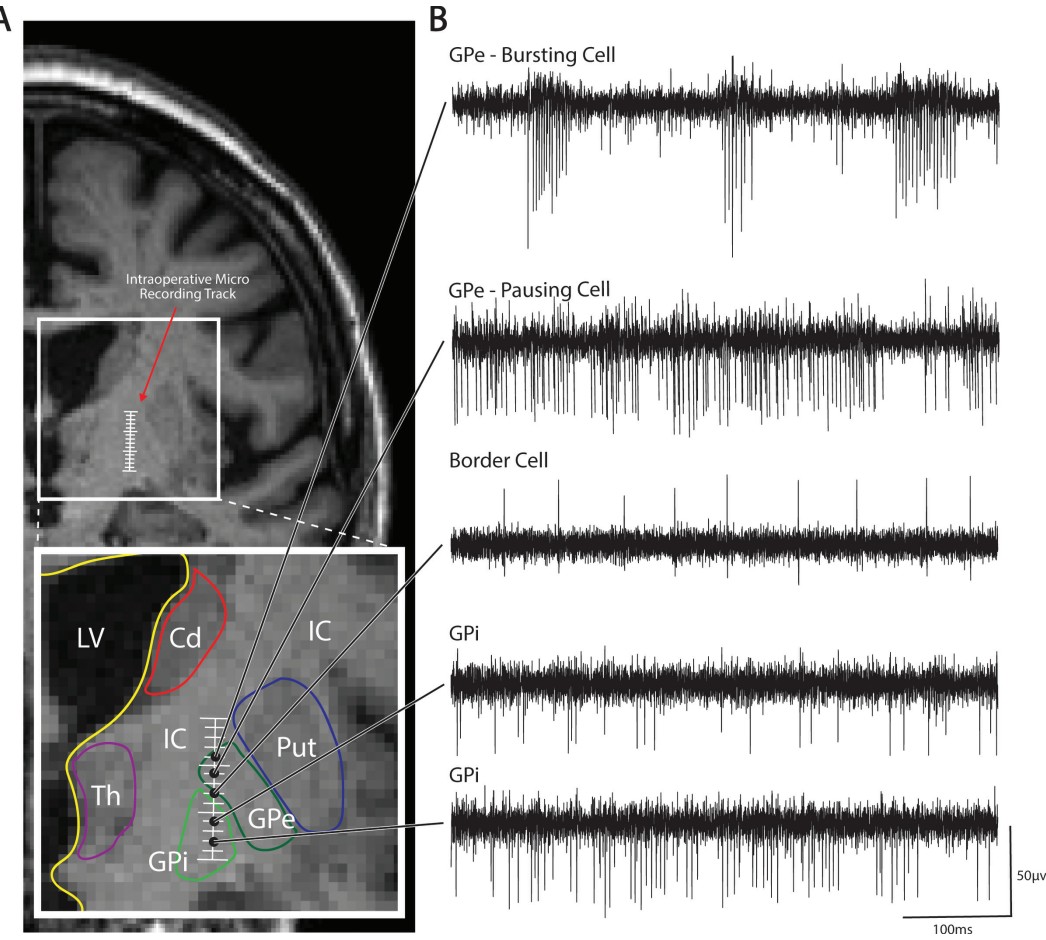

**Fig 2. Typical 'cell-types' seen on microelectrode recordings targeting the GPi. (A)** Representative pseudo-coronal T2 MRI displaying intraoperative micro recording track and zoomed-in segmentation of the GPi and surrounding anatomy. **(B)** Typical cell-types seen on microelectrode recordings targeting the GPi include bursting and pausing cells in the GPe and border cells and irregular firing neurons in the GPi. GPi = Internal globus pallidus; GPe = External globus pallidus; Put = Putamen; IC = internal capsule; Cd = Caudate; Th = Thalamus; LV = Lateral ventricle.

and broadband activity (1/f shape of the PSD) which reflects asynchronous population firing [18,19]. For detailed CRP methodology, see [3].

## Serial parameterization and spike sorting

One challenge in spike sorting is the occurrence of overlapping spikes, where action potentials from multiple neurons coincide in time, making it difficult to accurately identify and separate the signals from individual neurons [20]. Because each individual spike is parameterized by $\alpha C(t)$, a single unit can be subtracted from the raw voltage trace and spike sorted a second time using the same voltage threshold (Fig 4A–4D). We performed this serial parameterization and spike sorting for all MER sites with more than one sorted spike unit. We show an example MER site with two sorted units where serial parameterization and spike sorting increased the number of spikes detected by over 13% (Fig 4E–4F). Overlapping spikes are most common in extracellular recordings that contain multiple units with high firing rates. While this can separate spikes occurring in the same window, complete overlap at the peak

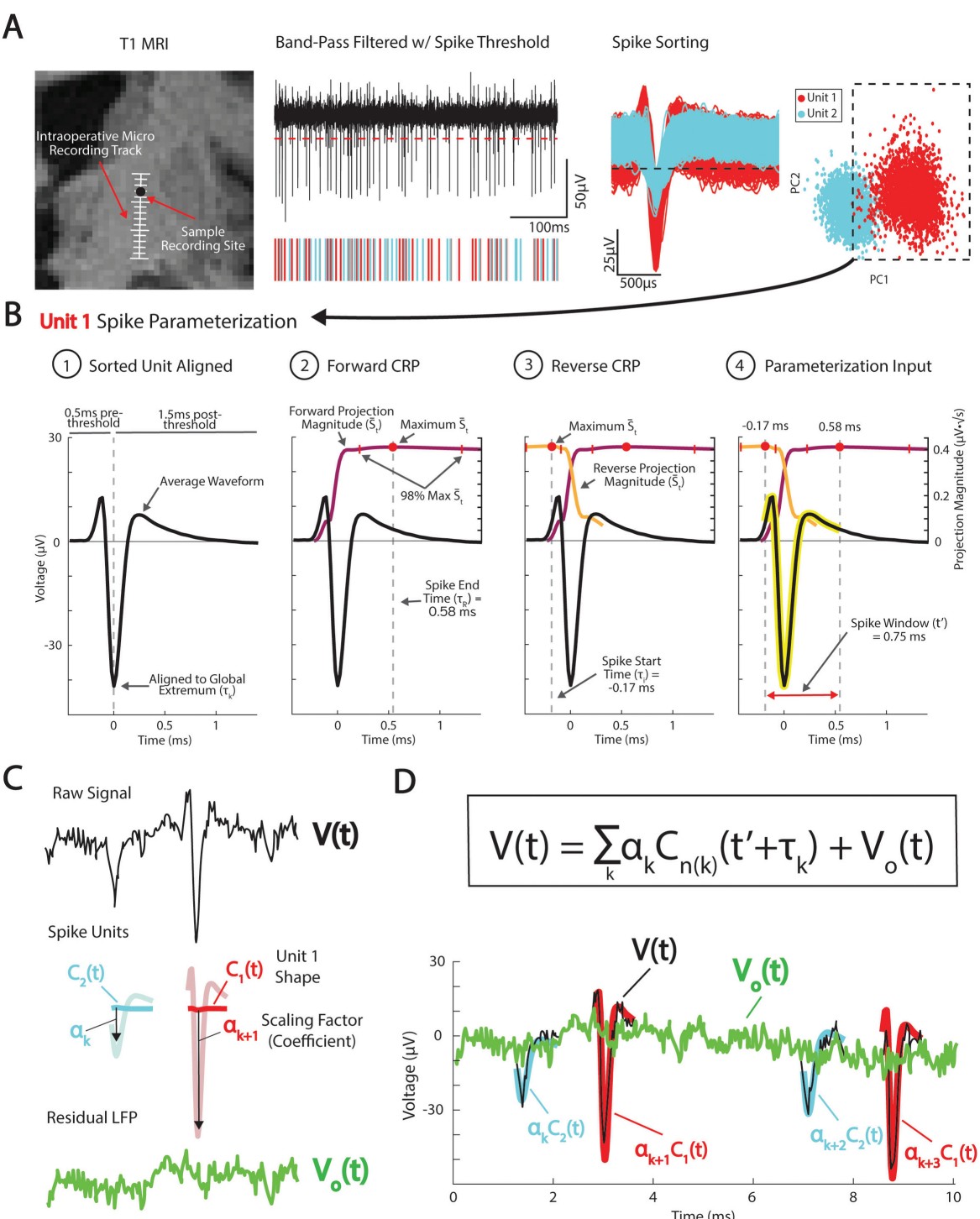

**Fig 3. Parameterizing spike units using CRP. (A)** Example of spike units sorted from a sample microelectrode recording site shown on pseudo-coronal T1 MRI. **(B)** Sorted units are parameterized using canonical response parameterization (CRP). Forward CRP identifies the spike end time point ($\tau_R$) at the maximum projection magnitude ($\overline{S}_t$). Reverse CRP starting from $\tau_R$ identifies the spike start time $\tau_i$ creating a significant spike voltage window. **(C)** Individual spikes are parameterized by sorted unit shape $C(t)$ and scaling factor $\alpha$. Removing scaled spike shapes from the raw voltage trace $V(t)$ leaves the residual local field potential $V_o(t)$ (LFP). **(D)** Overlaid $V(t)$ and $V_o(t)$ with $\alpha_k c_n(t)$, for spike $k$ of unit $n$, at spike times $\tau_k$ over duration $t'$: $V(t) = \sum_k \alpha_k C_{n(k)}(t' + \tau_k) + V_o(t)$.

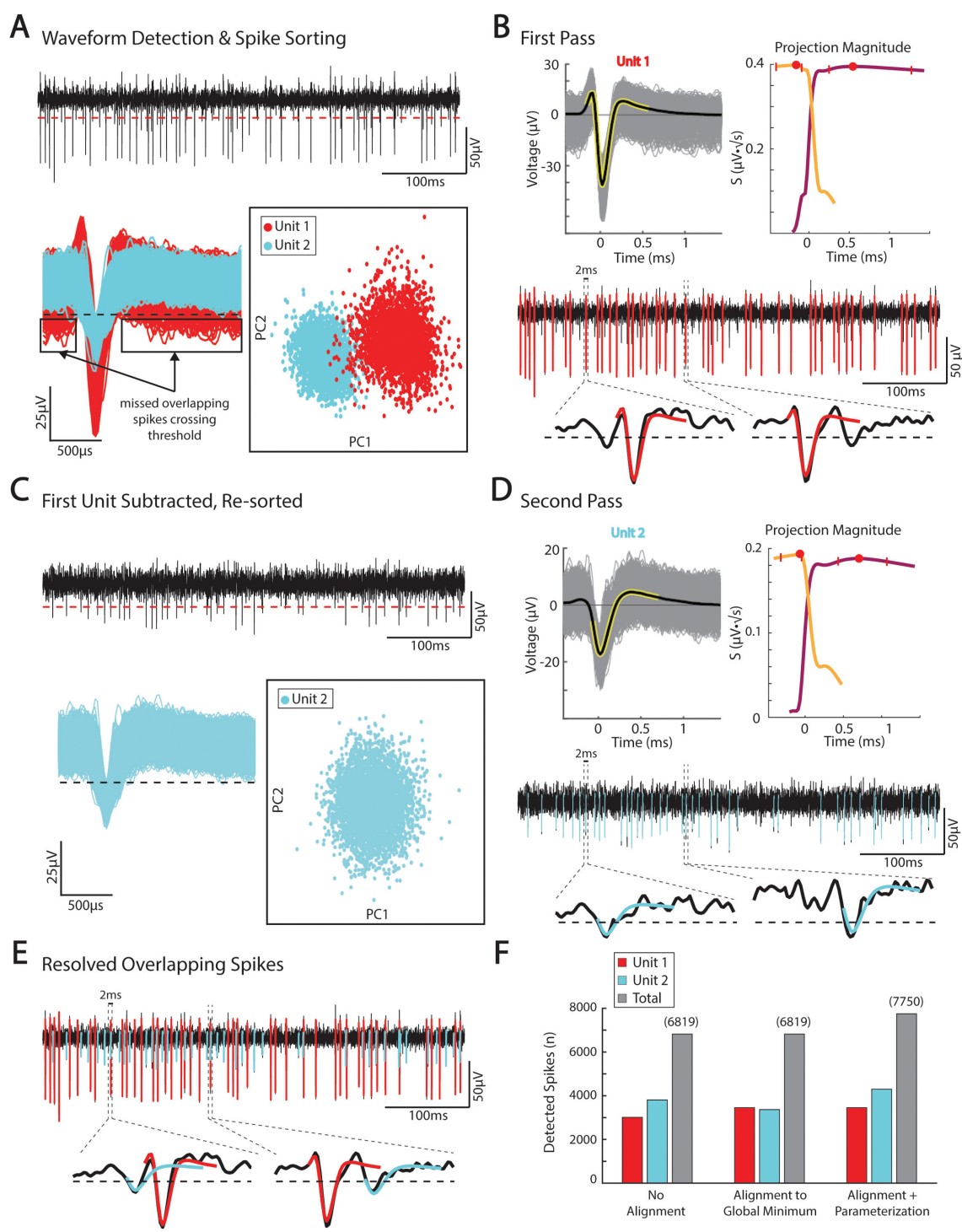

**Fig 4. Serial parameterization and spike sorting.** (**A**) Example spike sorted trace from Fig 3. Multiple spikes occurring in the same spike window are missed by standard spike sorting. (**B**) The largest unit is parameterized to obtain $\alpha_k C_{1(k)}$ for unit 1. (**C**) $\alpha_k C_{1(k)}$ is subtracted from the filtered voltage trace and re-sorted using the same voltage threshold. (**D**) The second unit is parameterized. (**E**) parameterization of both units shows resolved overlapping spikes. (**F**) For the sample voltage trace, serial parameterization and spike sorting increased the number of detected spikes by over 900.

cannot be resolved. 'Recovered' spikes assigned to a unit could belong to a separate unit, so refractory period violations and other sorting quality metrics should be reassessed after each sorting.

## Unit similarity and hierarchical clustering

To analyze the similarity of sorted unit shapes across recording sites, we define a custom similarity metric, which allows for direct comparison of different shapes. When two units with different spike time windows are compared, we selected the earliest spike start time $\tau_i$, and the latest spike stop time $\tau_R$ for a comparison window. The mean waveform for each unit is normalized (so that mean voltage = 0), and the similarity score is calculated by taking the dot product of the two normalized waveforms. This allows for a range of values from -1 to 1, with 0 indicating complete dissimilarity, 1 indicating identical shapes, and negative values indicating opposite polarity. This metric effectively captures the degree of overlap between shapes, irrespective of raw voltage values, which are largely attributable to the distance of recorded cells from the microelectrode tip. Using this method, a similarity matrix was created with all sorted units from one GPi DBS patient's MER track compared to all others (50 sorted units over 35 recording sites).

Hierarchical clustering was conducted, where the distance between two clusters was defined as the average distance between all pairs of elements in the two clusters. We used the agglomerative approach, which starts by treating individual spike units as a single cluster, then continuously combines them based on the selected similarity score threshold [21]. The distance matrix, which was derived by subtracting the similarity score from 1, served as the input for the clustering process. All negative values were set to a similarity score of 0. The hierarchical structure resulting from the clustering process was visualized using a dendrogram and the optimal number of clusters was selected graphically using the elbow method visualizing the intra-cluster distance score cutoff and the number of resulting clusters.

## Results and discussion

### Quantitative spike description, shape similarity, and hierarchical clustering

With the description $V(t) = \sum_k \alpha_k C_{n(k)}(t' + \tau_k) + V_o(t)$, several useful quantities for each spike $V(t)_k$ can be described: a "projection weight" $\alpha_k$; a scaled projection weight, $\alpha_k'$ that is normalized by the square root of the number of samples in $C(t)$; a scalar LFP summary term $\sqrt{V_o^T V_o}$ (magnitude of the LFP); a "signal-to-noise" ratio $\alpha_k/\sqrt{V_o^T V_o}$; the "explained variance" of the spikes is $1 - \frac{V_o^T V_o}{V^T V}$. These values are calculated for each spike, and can be averaged for each sorted unit to create summary metrics. Several examples of different unit shapes and their descriptive quantities are shown in Fig 5. Overall, This CRP approach for analyzing sorted units across serial recordings allows for empirical discovery of spike duration and structure (rather than assuming a pre-defined waveform duration). This technique allows for direct comparison of seemingly dissimilar unit shapes across recording sites. $\alpha'$ is normalized by the square root of the number of samples in C(t) (in $\tau_i$ to $\tau_r$ interval) and roughly captures the average voltage deflection from zero, comparable to the root-mean-squared metric. This is weighted only over the empirically-discovered significant spike interval, rather than a preselected waveform duration. notably, average voltage peak amplitude in extracellular recordings can largely be attributed to distance from the microelectrode tip and does not necessarily represent fundamental differences in cell-types or tissue architecture. In this

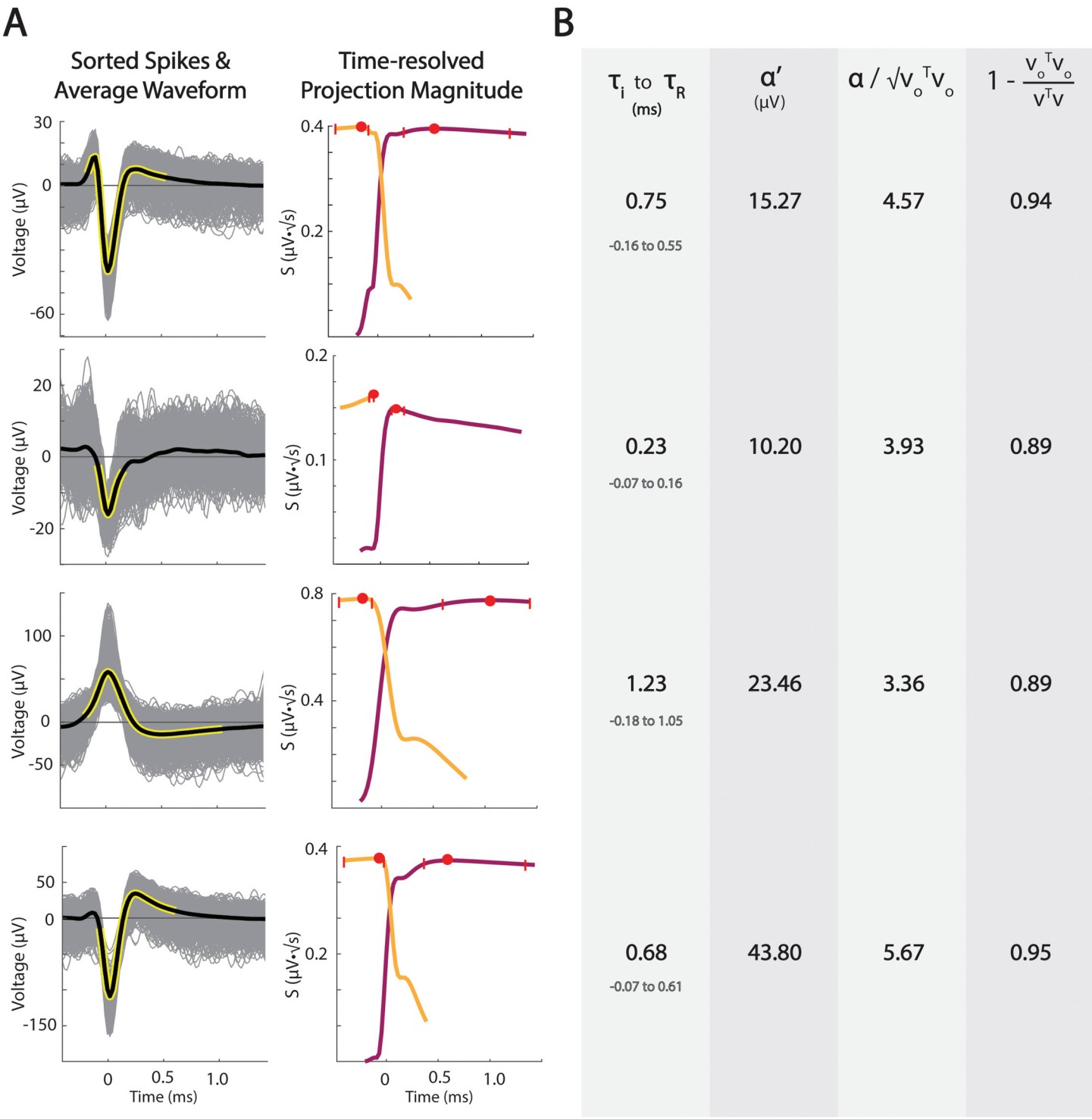

**Fig 5. Example of unit shapes, durations, projections, and parameterizations. (A)** Four examples illustrate different unit shapes, their durations, and projection magnitude profiles. **(B)** Parameterization values for example units, including spike window duration, spike magnitude, signal-to-noise ratio, and variance explained, respectively.

case, a more direct comparison of normalized action potential shapes may be needed for understanding the cellular diversity or identification of brain nuclei.

To capture unique action potential shapes, we employed a custom similarity metric between parameterized spike units. For this, a dot product is calculated between two normalized units from the outer $\tau_i$ to $\tau_R$. Spike similarity scores range from -1 to 1 with 1 indicating exact similarity, 0 indicating no similarity, and -1 indicating high similarity with opposite polarity. Fig 6 demonstrates 3 examples of highly similar units with different raw amplitudes (Fig 6A), units with inverse polarity (Fig 6B), and units with low similarity (Fig 6C). Spike similarity scores were calculated between all sorted spike units from one MER track (n = 50 sorted units from 35 recording sites). To perform hierarchical clustering, a distance score was calculated by subtracting the similarity score from 1. A dendrogram was created from the clustering and the elbow point method was used to determine the appropriate number of spike shapes (Fig 7A). Based on a distance score threshold at the elbow, we selected 6 unique clusters, producing resulting normalized shapes are shown in Fig 7C. Next we sought to determine whether sorted unit shapes were organized by depth. Spike shapes were plotted by the recording depth (Fig 8A). Surprisingly, unit shapes showed regional clustering in the globus pallidus, suggesting distinct neuronal types with conserved shape within anatomical

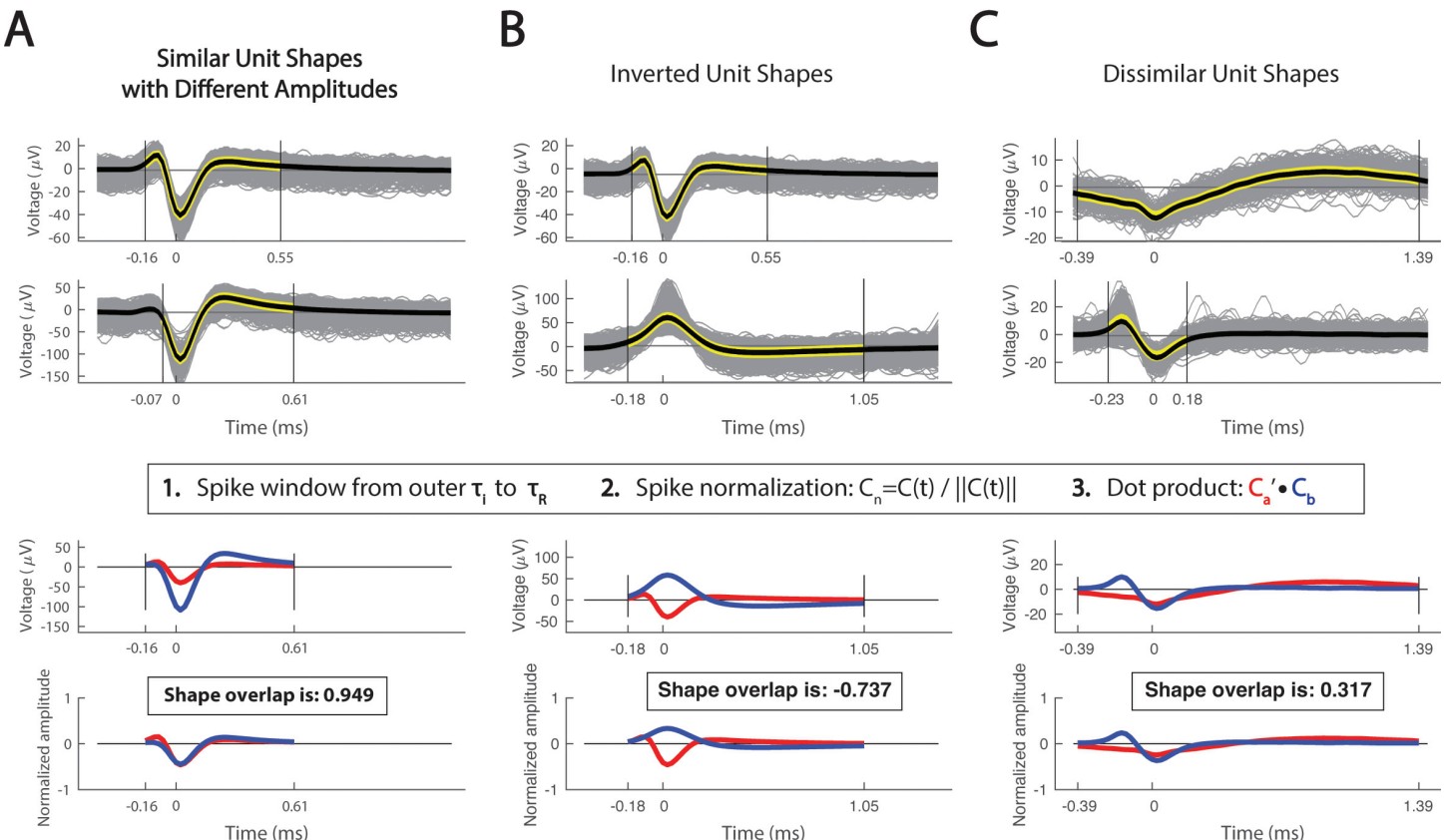

**Fig 6. Spike similarity comparisons across recording sites.** Spike similarity is calculated by taking the dot product of two amplitude-normalized units over the outer limits of $\tau_i$ to $\tau_R$. **(A)** Sample comparison of highly similar unit shapes with different raw amplitudes. **(B)** Sample comparison of two units with opposite polarity and a negative similarity score. **(C)** Sample comparison of two dissimilar unit shapes. A similarity score of 1 indicates identical shapes, -1 for identical inverted shapes, and 0 for completely dissimilar shapes.

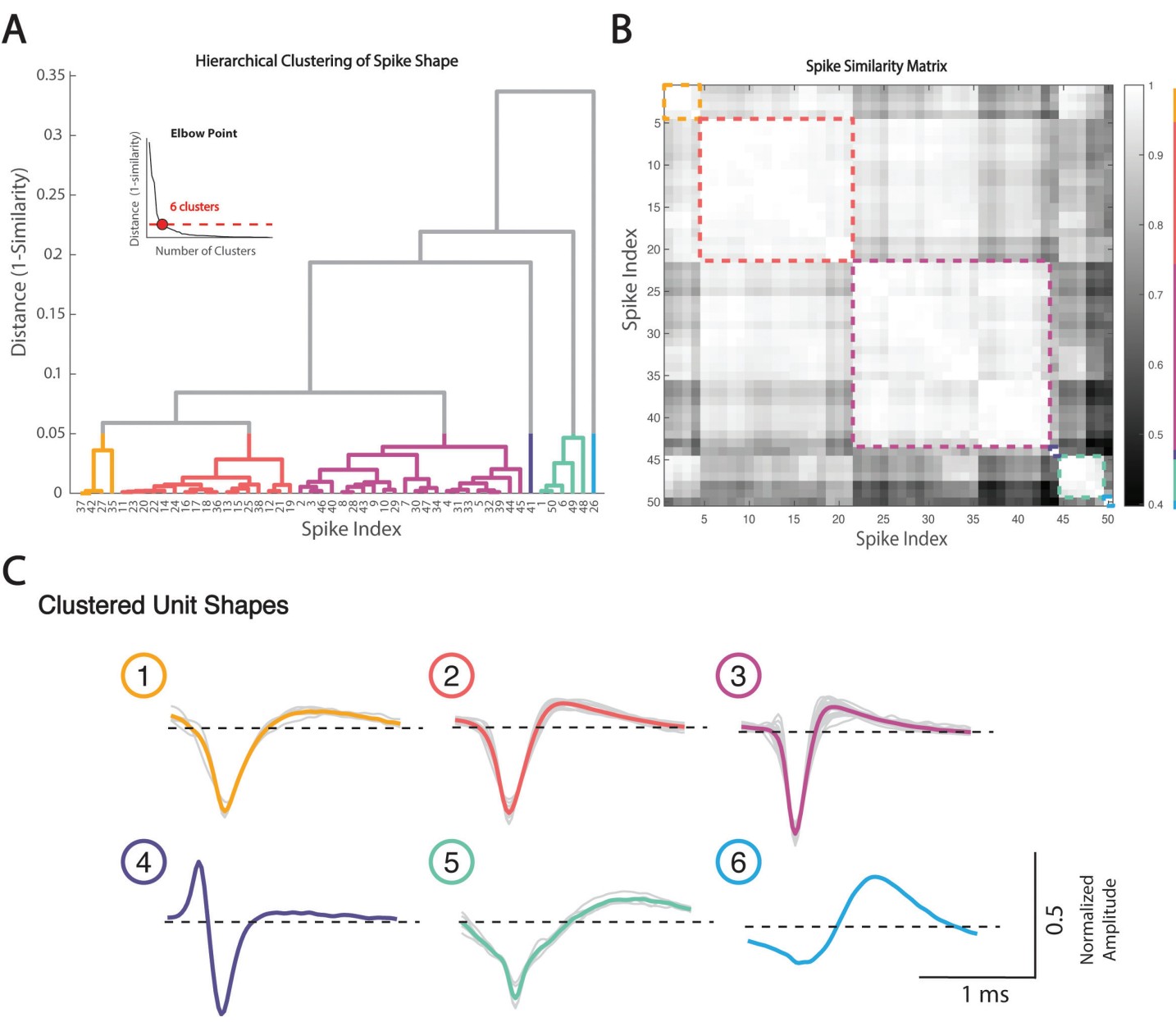

**Fig 7. Hierarchical clustering of spike unit shapes.** Hierarchical clustering was performed on the distance metric (1-similarity score) of all sorted units in the left GPi microelectrode recording track. **(A)** The hierarchical structure was visualized using a dendrogram. The number of unique cluster shapes was determined to be 6 based on the elbow point method. **(B)** Spike similarity matrix and resulting clusters. **(C)** Normalized unit waveforms and average cluster shape. Shape 3 (n = 22) and shape 2 (n = 17) were the most common shapes present, followed by shapes 5 (n = 5) and shape 1 (n = 4). Shapes 4 and 6 were both composed of one unit.

boundaries. Next, to learn if these depths were consistent with brain regions, we plotted spike identities along the MER physiology track on patient T2 and FGATIR MRI series (Fig 8b).

As shown in Fig 1e, a typical GPi MER track may travel through 2–3 neuronal structures including the GPi, GPe, and putamen (although putamen may not be present in the recordings from any particular patient due to the cannula opening lying below the transition into the GPe). The GPi is further divided into its own internal and external segments, which also

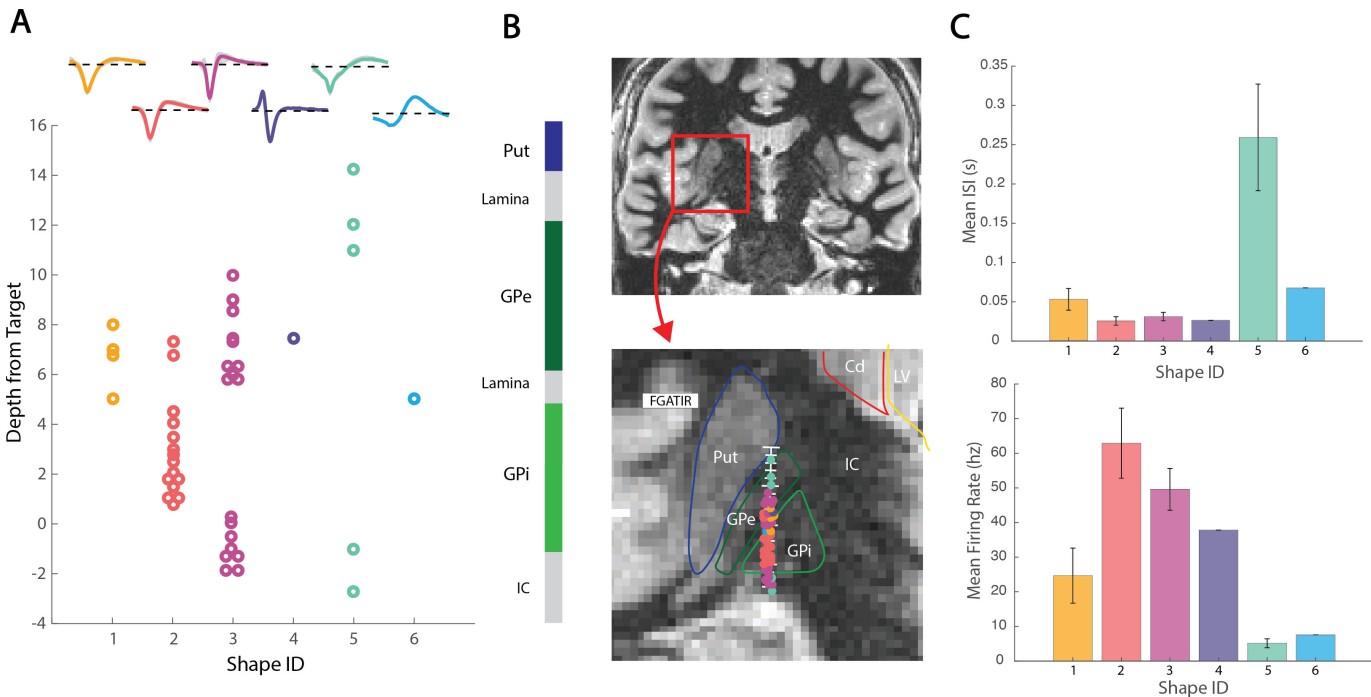

**Fig 8. Spike shape clustering on anatomy. (A)** The 6 clustered spike shapes plotted by depth from target in the GPi. **(B)** Spike shapes are plotted along the visualized microelectrode recording track on hand-segmented anatomy on psuedocoronal FGATIR MRI. Spike shapes showed regional clustering, consistent with white matter (lamina, IC), GPe, and GPi regions. **(C)** Mean (+/- SEM) inter-spike intervals and firing rates across clustered spike shapes. Shape 5 was present at the very top and very bottom of the track. Shape 3 was divided into 2 groups from +6 mm to +10 mm and -2 mm to 0 mm from target. Shape 2 was present primarily between +1 mm and +5 mm. Shape 1 was closely intermixed with the upper cluster of shape 3 units. GPi boundaries were closely associated with shape 2 and the bottom cluster of shape 3. Segmented GPe boundaries were associated with shape 1 and the top cluster of shape 3. Shape 5 occurred near the top and bottom of the MER track. Put = Putamen; GPe = External globus pallidus; GPi= Internal globus pallidus; IC = internal capsule.

show functional and anatomical differences important for DBS targeting [22,23]. Interestingly, we found 4-5 distinct patterns of spike shapes along the MER track, spatially correlating with expected and hand-segmented regions based on postoperative lead localization on imaging (Fig 8). Results for a second patient showed the same prevalent shapes and similar spans (S1 Fig).

**Comparison to conventional waveform clustering methods.** To compare our shape-based clustering approach with more conventional methods, we performed an unsupervised analysis in which mean spike waveforms were clustered using principal component analysis (PCA) followed by k-means clustering. PCA was applied using the full waveform as a feature vector, and k-means clustering was performed with six specified clusters. The Hungarian method, a cluster optimization algorithm, was used to quantify overlap [24]. When using raw waveforms, clustering results were dominated by amplitude differences, and cluster morphology diverged substantially from those derived via our CRP-based similarity metric (S2A Fig; 52%). After normalizing waveforms to remove amplitude as a confound, clustering results more closely resembled our shape-based clusters; however, agreement remained partial (S2B Fig; 66%).

## Biological interpretation

Rodent studies have significantly advanced our understanding of brain structure and function at single-neuron resolution which may inform putatively homologous regions in humans. The

GPe has been shown to be an important structure in the indirect basal ganglia pathway, regulating motor output [25]. Rodent studies show that the GPe is composed of primarily two types of GABAergic cells: prototypic and arkypallidal neurons [26,27]. Prototypic neurons make up approximately 2/3 of GPe neurons and project downstream to the STN and other output nuclei. Arkypallidal neurons comprise about 1/4 of GPe neurons and project back to the dorsal striatum (homologous to caudate and putamen in humans). Electrophysiology studies show differing firing and spike shape characteristics between prototypic and arkypallidal neurons, with prototypic neurons generally having a higher, more tonic firing rate, and a smaller spike width [28,29]. While we do not have sufficient power to compare spike firing properties from only one MER track, our discovered shape 3 in Fig 8 is more common (n= 9 vs 4), has a narrower spike width, and has a higher firing rate compared to shape 1 in the GPe segment, suggesting shapes 1 and 3 might represent prototypic and arkypallidal neurons, respectively.

The GPi is a part of the direct basal ganglia pathway and one of the main DBS targets for PD and dystonia [30]. The rodent homologue of the GPi is the endopeduncular nucleus (EPN)[31,32]. While neuronal sub-types are not as clearly distinct in the EPN/GPi as they are in the GPe, the GPi is similarly composed of inhibitory GABAergic projection neurons. And, although the main output related to motor function is the thalamus, there are also projections to the habenula which may regulate other non-motor functions [33]. Human and non-human primate studies have not investigated these specific projections, but have identified distinct functional subregions within the GPi, primarily along a dorsal/ventral axis [34]. Our discovered shapes 2 and 3 showed distinct spatial clustering from one another within the GPi, suggesting that each may reflect a characteristic cell type found within the internal and external GPi subdivisions (Fig 8). Surrounding these basal ganglial nuclei are white matter tracts and internal lamina which generally show either a very weak background of cells or an apparent absence of identifiable unit activity.

While comparison of these basic spike firing properties between clustered shapes are limited to a single patient for this methodological illustration, clustering of a large cohort of sorted units over multiple patients and tracks will allow for firing rate, ISI, and other firing pattern features to contribute to a more general statistically-powered description of characteristic cell-types related to anatomic subregions. Importantly, while spike morphology alone may be useful and actionable for understanding cellular diversity and classification in these regions, "ground truth" confirmation of genetically defined cell-types will require either comparative studies in animal models or post-mortem human tissue using genetic and histological tools.

## Neurosurgical applications

The identification of putative cell-types and brain regions based on spike shapes could have significant applications to neurosurgical practice in deep brain stimulation procedures. Initially, clearer and more precise intraoperative electrophysiological mapping using MER would lead to more accurate targeting and lead placement and improve therapeutic outcomes while minimizing off-target effects. As we begin to better understand the mechanisms of deep brain stimulation and movement disorders, optimal therapy could target stimulation of certain cell-types along the DBS lead trajectory, using shaped current or optogenetic pulses [35]. Compared to current standard MER practice, spike shape targeting may provide robust physiologic information beyond spike rates and firing patterns, which can significantly vary across subregions or external factors such as anesthesia drugs and disease types [11,12,36,37]. Therapeutically, these findings could be incorporated into intraoperative tools for automated

classification and identification of anatomical boundaries and subregions, implant targets, and other disease-related physiology.

## Other applications

**Modeling amplitude-modulated units from the cardiac cycle.** While Fig 5 shows summary metrics for parameterized spike units, individual spike metrics can also be examined over time to characterize how a unit is modulated by intrinsic or extrinsic factors like cardiac rhythm, neural oscillations, behavioral tuning, and others. Fig 9 shows an illustrative example of this, from a thalamic recording site, where spike amplitude was modulated by the cardiac

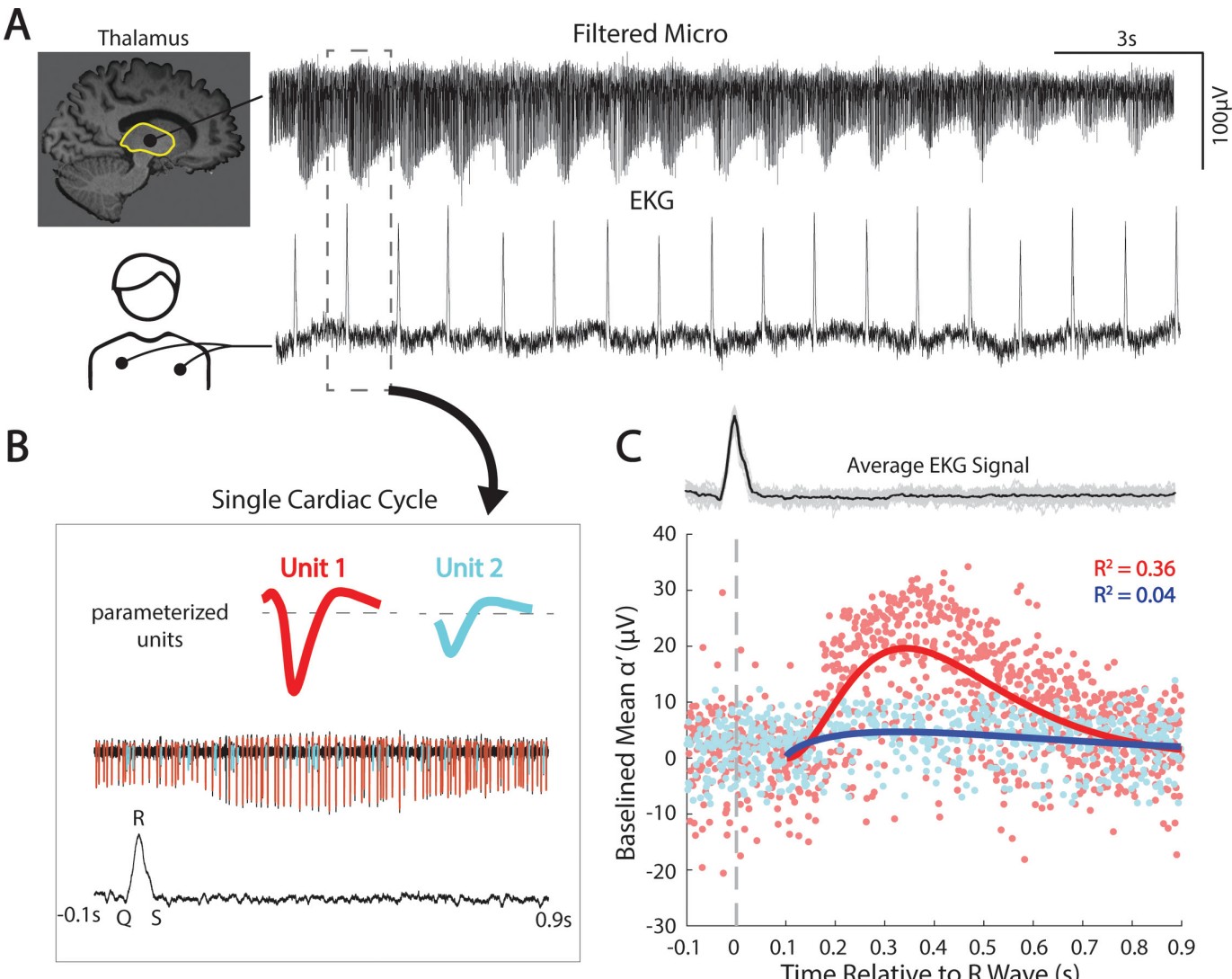

**Fig 9. Capturing pulse-modulated firing using $\alpha'$. (A)** Raw signal traces from the motor thalamus microelectrode recording and EKG. **(B)** a single cardiac cycle was isolated (-0.1 to 0.9 from R-wave) with two parameterized units. **(C)** The mean $\alpha'$ of each parameterized spike was plotted in reference to the aligned R-wave of the average EKG signal. An impulse-response function was fitted to the baselined distribution using a nonlinear least square fitting algorithm using the trust-region-reflective method in the form $At^n e^{-mt}$. Unit 1 shows clear impulse-response-like pulse modulation (A = $5.43e^3$; n = 2.33; m = 9.60; $R^2$ = 0.36), whereas uni2 did not show cardiac-tuning (A = 24.79; n = 0.67; m = -2.95; $R^2$ = 0.04). $R^2$ values show that unit 1 is pulse-modulated, while unit 2 is not.

cycle (based on measured electrocardiogram from the chest, Fig 9A). We sorted and parameterized two spike units, and windowed -0.1 seconds before and 0.9 seconds after each detected cardiac R-wave on EKG (Fig 9B). Then we averaged all cardiac cycles and plotted the $\alpha'$ as an impulse-response function of the cardiac window. We then utilized a nonlinear least square fitting algorithm using the trust-region-reflective method. $\alpha'$ as a function of time after the EKG "R" peak was fit to an impulse-response model form, $At^n e^{-mt}$. In the example, two units were isolated from the same trace, and, interestingly, one was strongly pulse-modulated and the other was not modulated at all (Fig 9), which can represent changes in physical distance to the microelectrode tip or possible engagement of mechanoreceptors sensitive to local blood pressure changes [38].

**Spike removal for spectral analyses.**   While the primary thrust of this methodological work is to show how spikes can be effectively isolated and parameterized, the technique is also useful because the residual LFP term, $V_o(t)$, represents all local brain activity aside from the action potentials, combined with measurement noise due to amplifiers and the operating room environment. The residual LFP can be isolated by subtracting $\alpha C(t)$ for all sorted units from the raw signal trace (Fig 10A–10D) and separately analyzed. For example, there is a clear decrease in spectral power above ~150 Hz when the spike shape contamination is removed. If the act of removing spikes introduced discontinuities (i.e. created artifactual contamination itself), this process of spike removal would instead result in an artificial *increase* in power. One important caveat, however, is that the phase response of the recording amplifier may not be flat across the frequency spectrum. This could introduce distortions into both the spike waveforms and the reconstructed residual signal, particularly in higher-frequency bands, and may limit the interpretability of subtle spectral features in this range. This spike removal prior to spectral analysis may be important when examining interactions between single units and background LFP oscillations, but also when studying so-called "broadband" or "1/$f$" phenomena, which are believed to reflect average synaptic activity [18,19] (Fig 10). Proper isolation of the LFP is critical for understanding larger population-level dynamics and how they relate to spiking data.

## Limitations

There are important limitations to consider with this approach. First, this study was conducted on densely sampled recordings from a small number of patients, which limits the generalizability of our findings. Future studies will assess group-level clustering and create classifier models to predict clinical-labeled boundaries from spike shapes and firing. While we observed distinct spike shapes across anatomical regions that were largely consistent in both patients, other factors beyond cell-type such as cellular orientation, glial interactions, and neuromodulatory tone may influence waveform structure. While our method normalizes unit amplitude to reduce the impact of electrode distance, any ground truth assessment of cell-type would require a mixture of transgenic animal models, optogenetics, histology, and others. Additionally, some spike shapes observed in white matter regions may reflect some combination of background activity and amplifier/background noise rather than discrete units and should be interpreted cautiously. Regardless, even reliably identifiable shapes from background spiking/noise could still prove useful for classifier models. While this approach offers advantages over existing techniques, such as empirical spike windowing and shape based similarity, it is designed to compliment, not replace, standard spike sorting and shape comparison methods. Although, principles could be integrated into automated pipelines to enhance spike detection and unit classification similar to other matching pursuit methods [39].

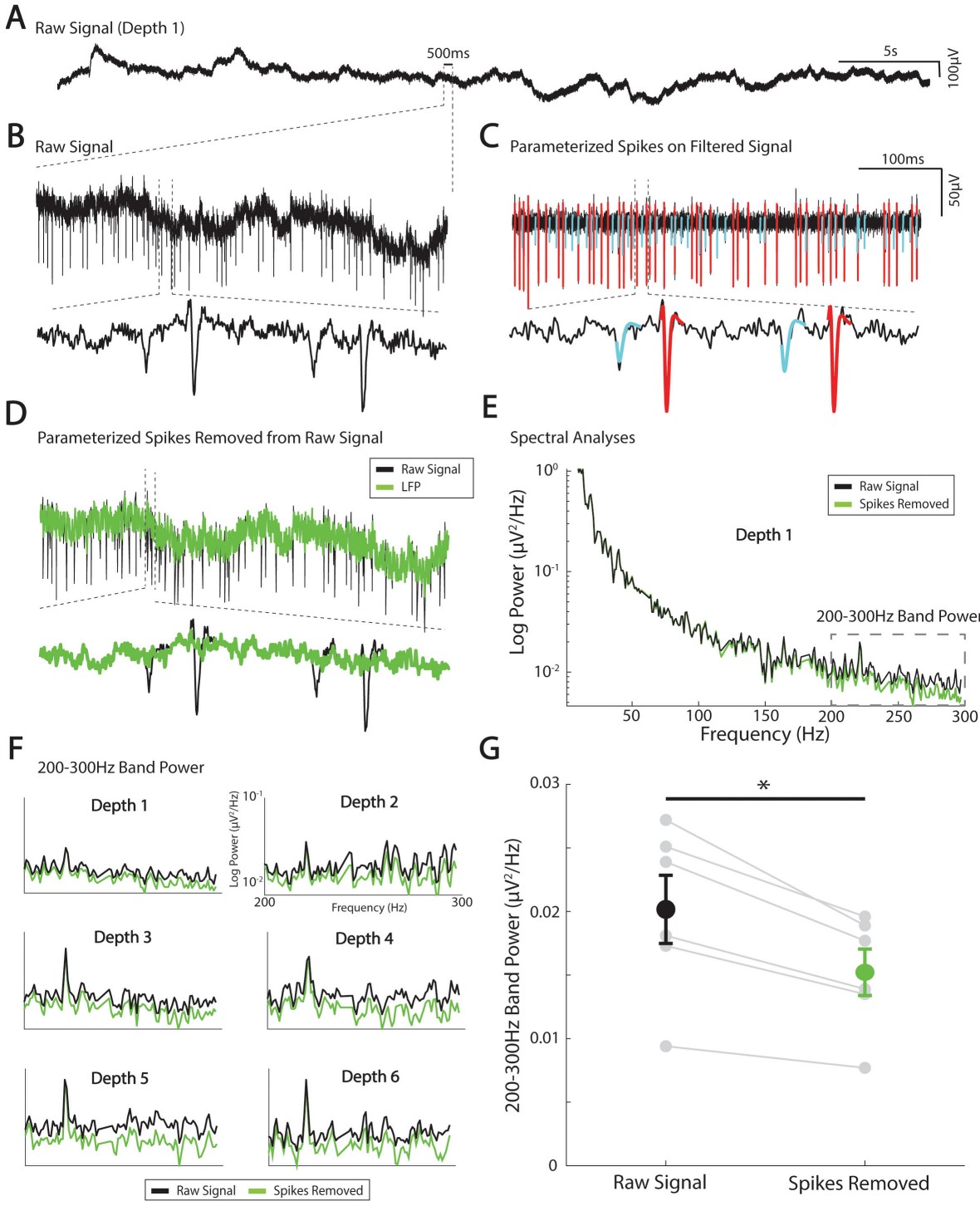

**Fig 10. Local field potential isolation using spike parameterization and removal. (A)** Example raw voltage trace from a microelectrode recording site. **(B)** zoomed in raw voltage trace. **(C)** Two sorted units parameterized and plotted on the filtered voltage trace. **(D)** subtracting the parameterized spikes from the unfiltered trace isolates the local field potential. **(E)** Power spectral density was calculated from the raw and LFP signal with spikes removed. **(F)** Average 200–300 Hz band power at 6 different recording sites from the raw signal (black) and LFP (Green). **(G)** Spike removal at 6 recording sites from one MER track showed to a significant reduction in 200–300 Hz band power by approximately 25% (paired t-test, t(5) = 5.37, *p = 0.003).

## Supporting information

**S1 Fig. Spike Shape Clustering for Patient 2. (A)** Normalized unit waveforms and average cluster shape for the 6 unique shapes identified from hierarchical clustering. Shapes 2, 3, and 4 are the most common shapes, consistent with figures 7-8. **(B)** The 6 clustered spike shapes plotted by depth from target (mm) in the internal globus pallidus (GPi). **(C)** Spike shapes are plotted along the visualized microelectrode recording track on hand-segmented anatomy on psuedocoronal FGATIR MRI. Average +/- SEM firing rate (D) and interspike interval **(E)** across shape clusters.
(TIF)

**S2 Fig. Comparative Clustering Methods.** We compared hierarchical clustering outcomes from Figs 7 and 8 to directly clustering spike-sorted mean waveforms through unsupervised PCA (2D) followed by K-means clustering (specified same number of clusters) and compared overlap using the Hungarian method, a clustering optimization algorithm. **(A)** Directly clustering mean unit waveforms shared 52% overlap with the hierarchical clustering method, and were heavily influenced by unit amplitude. **(B)** By first normalizing the mean unit waveforms prior to PCA and clustering, the spike shapes more closely resembled our clustering findings, but still only shared 66% overlap.
(TIF)

## Acknowledgments

We are grateful to the patients who volunteered their time to participate in this research and to the staff at St. Mary's Hospital.

## Author contributions

**Conceptualization:** Matthew R. Baker, Kai J. Miller.

**Data curation:** Matthew R. Baker, Bryan T. Klassen, Kai J. Miller.

**Formal analysis:** Matthew R. Baker, Hossein Heydari, Kai J. Miller.

**Funding acquisition:** Matthew R. Baker, Kai J. Miller.

**Investigation:** Matthew R. Baker, Kai J. Miller.

**Methodology:** Matthew R. Baker, Kai J. Miller.

**Project administration:** Matthew R. Baker, Kai J. Miller.

**Resources:** Matthew R. Baker, Kai J. Miller.

**Software:** Matthew R. Baker, Michael A. Jensen, Hossein Heydari, Kai J. Miller.

**Supervision:** Kai J. Miller.

**Validation:** Matthew R. Baker, Kai J. Miller.

**Visualization:** Matthew R. Baker, Bryan T. Klassen, Michael A. Jensen, Gabriela Ojeda Valencia, Hossein Heydari, Kai J. Miller.

**Writing – original draft:** Matthew R. Baker, Kai J. Miller.

**Writing – review & editing:** Matthew R. Baker, Bryan T. Klassen, Michael A. Jensen, Gabriela Ojeda Valencia, Hossein Heydari, Nuri F. Ince, Klaus R. Müller, Kai J. Miller.

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
