## [Decision Letter · Decision Letter 0]

20 Mar 2025

PCOMPBIOL-D-25-00122

Parameterization of intraoperative human microelectrode recordings: Linking action potential morphology to brain anatomy

PLOS Computational Biology

Dear Dr. Baker,

Thank you for submitting your manuscript to PLOS Computational Biology. After careful consideration, we feel that it has merit but does not fully meet PLOS Computational Biology's publication criteria as it currently stands. Therefore, we invite you to submit a revised version of the manuscript that addresses the points raised during the review process.

Please submit your revised manuscript within 60 days May 20 2025 11:59PM. If you will need more time than this to complete your revisions, please reply to this message or contact the journal office at ploscompbiol@plos.org. Please include the following items when submitting your revised manuscript:

We look forward to receiving your revised manuscript.

Kind regards,

Shlomi Haar, PhD

Academic Editor

PLOS Computational Biology

Marieke van Vugt

Section Editor

PLOS Computational Biology

**Journal Requirements:**

2) Thank you for including an Ethics Statement for your study. Please include:

i) A statement that formal consent was obtained (must state whether verbal/written).

Potential Copyright Issues:

i) Figures 1, and 9. Please confirm whether you drew the images / clip-art within the figure panels by hand. If you did not draw the images, please provide (a) a link to the source of the images or icons and their license / terms of use; or (b) written permission from the copyright holder to publish the images or icons under our CC BY 4.0 license. Alternatively, you may replace the images with open source alternatives. See these open source resources you may use to replace images / clip-art:

5) Please ensure that the funders and grant numbers match between the Financial Disclosure field and the Funding Information tab in your submission form. Note that the funders must be provided in the same order in both places as well. Currently, the grant number provided by " National Institutes of Health (NIH)" is not the same in both places.

**Reviewers' comments:**

Reviewer's Responses to Questions

Reviewer #1: This paper presents a novel algorithmic approach to parameterize intraoperative human microelectrode recordings (MERs) to improve deep brain stimulation (DBS) targeting. The method uses canonical response parameterization (CRP) to characterize the structure and duration of action potentials, allowing for the clustering of spike morphologies based on their similarity. By applying this technique to DBS recordings, the authors found that spike shapes clustered regionally in the human globus pallidus, suggesting anatomical correlations. This approach enhances the precision of DBS lead placement and holds potential for broader applications in neurosurgical procedures.

This is a wonderful paper, with a refreshing take on spike analysis, following their already impressive work on CCEPs. However, there are a few areas where the manuscript could be improved.

The dataset used is small, consisting of only 3 patients. This is surprising, given that Mayo is such a clinically busy hospital. Including more patients, or using any of the various open-access MER datasets available on DABI or elsewhere, would help ensure the generalizability of this method.

There is no limitations section in the paper, which should be added. For instance, the speculation about cell-type-specific waveforms is not based on any ground truth analysis. There is no histology or other method to prove this association, and it’s certainly possible that other factors influence the recorded AP shape, such as the density of glial cells, the orientation of the cell, or neuromodulatory tone (from other projections). I still think the authors are definitely on to something, and this is a valuable analysis, but they should acknowledge that it’s not foolproof.

Could the authors comment on how enriching these data with more contextual information could provide additional insights? For example, a more sophisticated model that includes the presumed anatomical region, the interspike interval of detected units, etc., may be able to constrain the classification of imputed neuron clusters.

I may have missed this, but would it be possible, in principle, to reassess a recording with the identified putative neuron AP shapes as a means of improved spike detection—perhaps detecting smaller AP waveform amplitudes and enriching the number of detected units?

The authors note that detected spikes can be regressed out of the ongoing time series to provide an estimate of the LFP. Could they comment on how the recording amplifier’s phase response (which is likely not flat across the frequency spectrum) may or may not lead to some distortions here?

Reviewer #2: In this manuscript, Barker and colleagues introduce an approach for parameterizing the morphology of extracellular action potential recordings captured from the human brain. Briefly, the authors examine micro-electrode recordings of single units, and specifically the shape of the single units, in order to parameterize their structure and their duration of activity. They use this parameterization to hierarchically group spike shapes into clusters, and conclude by examining the relation between these clusters and anatomy. The main contribution here is that the authors use an approach built upon canonical response parameterization. In effect, this allows one to identify the precise duration of a single action potential so that the waveform between those time points can be compared to other action potentials in the recorded trace. By adopting this approach, for every single spiking event, the authors can extract out the specific duration of that spike, the scaling factor or amplitude (alpha) required to match that event to the template for that unit, and the residual LFP trace. This is an interesting approach, and could certainly be helpful for researchers focused on single unit spiking activity. However, there are some suggestions that could strengthen the overall conclusions of the work.

The study presented here appears to be largely a methodological report. How and why this could offer advantages over other approaches, however, is not clear. For example, if one were to simply group spiking events based on the template waveform extracted through spike sorting, would that yield results that are inferior to those presented here. In general, the paper presents an approach for extracting these features, but it is not clear how or why this approach should be better suited to human data than other approaches that are more routinely used in the literature.

One area where this could potentially be improved is in describing the results of this approach on neural data. However, here, the data and results presented are somewhat sparse. As far as I can tell, the hierarchical clustering result demonstrating the different anatomic correlates of different waveform clusters are all extracted from a single recording track in a single patient. The other main examples demonstrating how this approach can be helpful are correlating scaling factors with cardiac rhythms and extracting the residual LFP. These also appear to be single examples. The conclusions and the utility of this approach could be much improved if these analyses were applied across a larger set of subjects.

Reviewer #3: Baker et al. present an intriguing spike sorting and parameterization method applicable to DBS MER. This technique potentially offers an objective characterization of a process traditionally reliant on clinical expertise. However, several clarifications are necessary to fully interpret the study's results:

1) The authors mention obtaining recordings with the patient at rest. It is essential to clarify whether recordings involving somatosensory evoked potential testing or visual evoked potential testing were excluded. Additionally, were microstimulation or macrostimulation recordings included in the dataset? Were there any adjustments to the final DBS trajectory post-MER, ensuring that the DBS lead localization accurately represents the initial MER trajectory? Lastly, were any anesthetic or CNS sedating agents administered during these cases?

2) Can the authors provide information on the background and expertise of the individual who conducted the manual segmentation of the basal ganglia?

3) The study describes two GPi and one VIM case used to develop the technique. However, results are reported for only one GPi and one VIM case. Was the technique equally successful in the second GPi case?

4) The authors identified four distinct morphological populations within the white matter tracts surrounding the basal ganglia. Can they comment on the degree of heterogeneity suggested by this method? The Biological Interpretation section mentions an "apparent absence of identifiable unit activity" in these areas. Does this indicate true cells or noise? If it is noise, can this technique be used to identify it in an automated fashion?

5) The use of the term "local field potential" should be clarified, as LFPs are traditionally considered slow propagations (less than 100 Hz). Would LFP isolation, as demonstrated in Figure 10, applied to the 1-100 Hz band range, also show notable separation? In the 200-300 Hz range, does this represent the separation of spikes from ambient noise rather than LFPs?

**Have the authors made all data and (if applicable) computational code underlying the findings in their manuscript fully available?**

Reviewer #1: Yes

Reviewer #2: None

Reviewer #3: None

PLOS authors have the option to publish the peer review history of their article (what does this mean?). If published, this will include your full peer review and any attached files.

Reviewer #1: No

Reviewer #2: No

Reviewer #3: No

**Figure resubmission:**
---

## [Decision Letter · Decision Letter 1]

30 May 2025

Dear Dr. Baker,

We are pleased to inform you that your manuscript 'Parameterization of intraoperative human microelectrode recordings: Linking action potential morphology to brain anatomy' has been provisionally accepted for publication in PLOS Computational Biology.

Best regards,

Shlomi Haar, PhD

Academic Editor

PLOS Computational Biology

Marieke van Vugt

Section Editor

PLOS Computational Biology

Reviewer's Responses to Questions

**Comments to the Authors:**

Reviewer #1: The authors have addressed my concerns.

Reviewer #2: The authors have submitted a revision to address the first round of reviews. There were two major comments that were raised in the initial review. First, that this represents a relatively small sample size and therefore it is unclear how these results may be generalized. In response to this comment, the authors have noted that this is a limitation of this study, and that the results and data presented here are intended to introduce a methodological approach without any formal analyses to validate this approach. Similarly, the second major concern is how this approach may be more effective than standard approaches used to sort spiking data and to analyze neural responses. Here again the authors note that they are only introducing a methodological approach that can be used to complement available approaches for spike analysis. They are making no claims regarding whether one approach may be more effective or better than the other, and no claims regarding generalizability.

In this sense, if this were considered as simply a manuscript that introduces a new method that may or may not be generalizable and useful for subsequent analyses, then the authors have sufficiently addressed the concerns raised by the reviewers. This would be a nice additional approach to introduce to the field, and time will tell whether this will be an approach that is indeed adopted for neural analyses. If on the other hand the authors are making a claim that this approach could be more effective, or even useful for analyzing neural data, then the revision has not addressed those concerns. I do not believe that this is what the authors have intended, and so the current response and iteration of the manuscript appears sufficient for their stated limited goals. But I will defer to the editor and to the other reviewers regarding those broader claims.

Reviewer #3: The authors have adequately responded to my comments and concerns.

**Have the authors made all data and (if applicable) computational code underlying the findings in their manuscript fully available?**

Reviewer #1: None

Reviewer #2: Yes

Reviewer #3: None

PLOS authors have the option to publish the peer review history of their article (what does this mean?). If published, this will include your full peer review and any attached files.

Reviewer #1: **Yes: **John D. Rolston

Reviewer #2: No

Reviewer #3: No

---

## [Editor Report · Acceptance letter]

PCOMPBIOL-D-25-00122R1

Parameterization of intraoperative human microelectrode recordings: Linking action potential morphology to brain anatomy

Dear Dr Baker,

I am pleased to inform you that your manuscript has been formally accepted for publication in PLOS Computational Biology. Your manuscript is now with our production department and you will be notified of the publication date in due course.

With kind regards,

Zsofia Freund
